# Simultaneous costs minimizing in electricity and gas micro-grids with the presence of distributed generation

**Shahryar Behnia, Saeed Kharrati** \*, **Farshad Khosravi, Abdollah Rastgou**

Department of Electrical Engineering, Kermanshah Branch, Islamic Azad University, Kermanshah, Iran

\* dr.kharati@gmail.com

## Abstract

Distributed generation can actively participate in the day-ahead markets, real-time power balance, and wholesale gas markets to achieve various goals, such as supplying gas to various electric power generation plants. A multi-objective network with two types of loads is considered in this paper. The reason for the simultaneous optimization of these two networks is that these two energy carriers are dependent on each other and gas is needed to produce electricity, so this issue can be addressed with a multi-objective function. The simulation carried out in this article is coded in GAMS software as a mixed integer linear programming (MILP). The efficiency of gas turbines and fuel cells in this article is dependent on their working point, and considering the exact model of these resources and the relationships related to the calculation of their fuel consumption is non-linear. On the other hand, a binary variable has been used to show the charging and discharging state of the storage and the on-and-off state of the gas turbines. Therefore, the problem considered in this article is a MILP problem. The results of this article are the proper planning of charging and discharging of the energy storage system with the proper planning of the power generation of different energy sources considering the network loads in two optimized and non-optimized scenarios.

## 1. Introduction

Undoubtedly, one of the main challenges in the last century is the energy challenge. This challenge in various fields of energy, including energy supply, exchange and consumption, due to population growth, increasing global demand for energy, lack of fossil fuels and environmental pollution, and concern about energy security for all countries of the world, has become an important topic. The close relationship of energy with economy, social, environmental and security aspects has doubled its importance. Energy-related issues are important issues in this field, for example, we can refer to issues such as energy security, fair distribution, affordability, reliability, efficiency, and suitability for the environment [1].

Producing electricity near the place of consumption, in addition to reducing losses in the system, can have more flexibility to provide various services to consumers. With the increasing

**Funding:** The author(s) received no specific funding for this work.

**Competing interests:** The authors have declared that no competing interests exist.

penetration of distributed generation (DG) in power grids and considering different technologies, the uncertainties of its renewable types, Major challenges are raised for the design of power systems in the future. One of the methods of aggregating distributed generation resources is a concept called micro-grid (MG). MGs are low-voltage and medium-voltage active distribution networks that consist of a set of loads, distributed generation sources and control devices. Due to the independence in operation, these networks can be separated from the main network and operated as islands. The main purpose of forming these networks is technical issues related to increasing the reliability and quality of electricity delivered to consumers. Therefore, from the point of view of the upstream networks, the MG is an element with dual characteristics, which is sometimes considered a burden and sometimes a source of production.

With the significant growth of smart grids as well as the rapid improvement of communication platforms between production and consumption, MGs have moved towards smart-grid and the practical concept of electric smart MGs has been created. Using this model in case of adopting a comprehensive telecommunication infrastructure for energy management and protection is considered a suitable approach that can be applied to the current distribution networks in which the penetration of distributed generation will increase over time. Similar to the path of evolution of ordinary electric MGs to smart electric MG s and then the comprehensive management of smart electric MGs, Hub Energy has also evolved in the form of Hub Energy MGs, Smart Energy Hub MGs and a comprehensive network of Smart Energy Hub MGs [2].

In the past, different energy systems were planned and managed independently. But today, the development of technologies such as the efficient multi-generation system leads to the realization of the benefits of integrated energy infrastructures such as electricity, natural gas and district heating (DH) networks, and as a result, the rapid movement towards multi-energy systems (MES). In such systems, energy carriers and different systems interact synergistically. However, paying attention to such a concept requires a suitable tool for the integrated management of system components. An energy Hub (EH), which can be defined as a place where the production, conversion, storage and consumption of different energy carriers takes place, is a promising option for the integrated management of multi-energy systems.

According to reference [2], a centralized system of smart energy hub MGs is connected to a large central supplier. In this reference, the real-time pricing of the connected network is examined. Also, two types of smart hub MGs have been considered for connected centralized energy systems. Not using normal MGs, not checking distances and the possibility of connecting the introduced hub MGs in terms of distance, etc. can be considered as weak points of this research.

In reference [3–5], a transmission network of energy carriers in the form of gas and electricity is introduced, which is intelligently provided through one or two central suppliers and with a telecommunication connection, MG. The smart hub is fed in different areas. The one-way energy path from the supplier to the consumer, the lack of attention to distributed generation sources and smart energy hubs, and as a result the lack of two-way energy exchange between smart energy hubs, are among the weak points of this research.

According to reference [6], an advanced hybrid power generation cycle was evaluated to obtain sustainable energy with high power and efficiency. This combined cycle includes biomass gasification, Cascaded Humidified Advanced Turbine (CHAT) and steam turbine. The fuel consumed by the system is obtained from the gas produced in the process of conversion to biomass gas.

Reference [7] focuses on minimizing operational cost and optimal power distribution associated with MGs coupled with natural gas networks using Particle Swarm Optimization (PSO). Introducing a natural gas turbine in an MG to overcome the disadvantages of renewable

energy sources is a recent trend. This increases the load and congestion in the gas network. To avoid congestion and balance the load, it is necessary to coordinate with the power grid to plan for the optimal transmission of both interactive networks.

Reference [8] presents a comparative case study with multiple biomass sources to analyze biological feedstock logistics plans with distributed warehouses and a primary warehouse located with the biological. An MILP model for simultaneous optimization of feedstock decisions and optimal preprocessing warehouse locations and sizes, using biomass sources from agricultural residues, energy, and municipal solid waste to meet carbohydrate specifications and feedstock demand for a biochemical conversion process is developed.

Reference [9] proposes a multi-product multi-source framework for multi-carrier energy sources coupled with a biogas-solar-wind hybrid renewable system. In this framework, biogas-solar-wind complements are fully exploited based on the thermodynamic effects of digestion for the synergistic interactions of electricity, gas and heating energy flows, and a coupling matrix is formulated to model production, conversion, storage and consumption. According to reference [10], a practical method for analyzing the economic distribution of a power system is described using the Lagrange coefficient method according to the Cohen-Tucker conditions, without considering the limitations and transmission losses.

Reference [11] provides a new perspective and a complete mathematical tool to study the integrated energy system from the illegal market perspective. A Mathematical Program with an Equilibrium Constraint (MPEC) model is proposed to study the strategic behaviors of a profit-oriented energy pole in the electricity market and the heating market under the background of energy system integration. At the top level, the Energy Hub provides prices and quantities to an electricity distribution market and a heating market. At the lower level, two markets are settled and energy contracts are determined between the energy hub and two energy markets. The network limitations of physical systems are explicitly shown by an optimal power flow problem and an optimal heat flow problem.

Reference [12] presents the coordinated energy management of energy hubs in different networks based on the cooperation of hubs in Day-ahead markets. The proposed model connects electricity, natural gas and district heating networks by considering electricity, and natural gas as hub input and electric energy and district heating as hub output.

Reference [13] focuses on the optimal scheduling of regional energy systems with multiple energy supply modes and flexible loads. For the multi-energy system, the energy hub model including an energy storage system and integrated electric vehicle is developed.

Multi-Carrier Hub Energy System (MCHES) meets different energy needs such as heating, cooling and energy demand by using different energy sources simultaneously. According to reference [14], a robust optimization approach (ROA) for robust multi-carrier energy hub system planning considering economic and environmental constraints in the presence of market price uncertainty and multi-demand response programs (DRP) is provided.

In reference [15], there is a review of gas and electricity routes at the same time. This reference is a review of research considering the balance of these two energy carriers to become an energy hub and the works of this energy field have been reviewed. Reference [16] has also measured the effect of electrical routes on decision-making about gas routes and vice versa and has optimized these routes accordingly. In reference [17], the simultaneous effect of these two networks in macro-decision-making to become an energy hub has been considered.

The expansion of renewable production sources that are distributed in the network creates many challenges for the electricity distribution networks. Recent advances in battery technologies have made battery energy storage systems (BESS) more economical than in the past, which is why these equipment are used. It is suitable for use in the network and at the distribution level. Also, the distributed BESS can act as a factor for more penetration of renewable

energies in the distribution network [18]. This reference obtains the most optimal mode for the placement and size of batteries. This situation is obtained according to the cost of batteries, operation, maintenance, etc.

According to reference [19], each MG operator communicates with the distribution network operator (DNO) in the control layer to manage consumption and production. DNO plans production resources by considering the possible islanding of MGs in the distribution network. In reference [20], a sensitivity analysis is performed to investigate the impact of these technologies on unbalanced low-voltage residential networks located in Northern Ireland. The influence of different powers has been investigated and the results have been evaluated using different technical indicators. In reference [21], an optimal technique for the optimal location of the battery energy storage system (BESS) and the wind power penetration capacity for battery charging/discharging is proposed.

Reference [22] deals with the optimal allocation of BESS in radial distribution systems to regulate bus voltage and reduce energy costs. According to reference [23], in recent years, the BESS has been considered a promising solution to reduce wind turbine generator frequencies.

In references [24,25], the improvement of wind turbine performance to reduce the fluctuation of output power and also to increase the extracted energy in an MG has been investigated. Also, in references [26–28], the lifetime estimation of electrical network equipment has been reviewed.

In distribution networks, the voltage stability index is an important issue and is considered a security objective function. Therefore, it is necessary to check this index by energy management in the distribution network. According to [29], this index is taken into account and dynamic distribution feeder reconfiguration (DDFR) is introduced as a suitable method for this management. This study has been considered by considering the voltage stability index, power loss and operation cost in a distribution network with the presence of PV, energy storage system and capacitors.

In distribution networks, electricity prices and load patterns are constantly changing, and accordingly, many operational problems will affect electricity distribution networks. Therefore, to prevent these incidents, the problem will be considered in different time frames. In [30], a multi-objective optimization model is presented using the DDFR procedure in the distribution system and in multiple time intervals with the presence of energy storage systems and PV units. In [31], a cost-effective energy hub system is proposed using linear programming, considering transformer converters, combined heat and power (CHP), heat exchangers, and energy storage, including electric vehicle charging stations (EVCSs), and heat storage.

Market exchanges for placement and appropriate size of DGs in an electricity distribution network have been investigated by proposing an appropriate algorithm in [32]. Also, the energy management of the MG connected to the grid is described in [33].

The short-term planning of the network to use renewable energy sources and increase their penetration has also been evaluated in [34]. Electric energy management in the smart rural network has also been investigated according to [35].

By reviewing the references and stating the goals of each one, the summary of the work done and its comparison with the innovations of this paper is shown in Table 1. According to this table, it is clear that the simultaneous examination of gas and electricity in the network, considering different types of load to analyze the results, and considering the energy hub management are among the innovations of this article that are less discussed in previous works. Also, the type of objective function in this work is one of its other specific innovations.

In the basic state, MGs work as islands or not islands (connect to the grid). But the main point is the optimal communication of these MGs and their intelligent management and exploitation; in a way that can achieve the most optimal mode of using energy from a technical

**Table 1.** *Compare the proposed method and previous methods.*

| No. | Aspect | [19] | [20] | [21] | [22] | [23] | [24] | [25] | [29] | [30] | [31] | This paper |
|---|---|---|---|---|---|---|---|---|---|---|---|---|
| 1 | Energy Losses | | ✓ | ✓ | | ✓ | | | ✓ | ✓ | | |
| 2 | Prediction Model | | | | | ✓ | ✓ | ✓ | | | | |
| 3 | EV Considering | | | | | | | | | | ✓ | |
| 4 | Unbalance Load | ✓ | ✓ | ✓ | | ✓ | | | | | | |
| 5 | Demand Response Considering | | | | | | | | ✓ | ✓ | ✓ | |
| 6 | Feeder Reconfiguration | | | | | | | | ✓ | ✓ | | |
| 7 | Voltage Stability Considering | | | | | | | | ✓ | ✓ | | |
| 8 | ES & different DGs Considering | | ✓ | ✓ | ✓ | ✓ | ✓ | ✓ | ✓ | ✓ | ✓ | ✓ |
| 9 | Multi-Objective Analysis | ✓ | ✓ | ✓ | ✓ | ✓ | | | ✓ | ✓ | ✓ | ✓ |
| 10 | 24-Hours Analysis | ✓ | ✓ | ✓ | ✓ | ✓ | | | ✓ | ✓ | ✓ | ✓ |
| 11 | EH Managing | | | | | | | | | | ✓ | ✓ |
| 12 | Load Types Considering | | | | | | | | | | | ✓ |
| 13 | Simultaneous gas & Electricity Considering | | | | | | | | | | ✓ | ✓ |

– EV: Electrical Vehicle–ES: Energy Storage–DG: Distributed Generation–EH: Energy Hub.

and economic point of view. Therefore, in this article, we will try to optimize the objective function with two issues electricity and gas. The purpose of optimization is to reduce network costs. In the non-optimized state, the costs of both energy sources will be much higher and we will not reach the optimal point in fuel consumption. In other words, we may go to gas power plants during the hours when we need electric energy and we can use relatively cheaper energy such as solar energy during the day, thus increasing the network costs. However, with optimal energy consumption and production side management, network costs can be optimized and this cost can be transferred to investment in other sectors. This review will consider the 24-hour optimal points for the MG from the perspective of cost.

Based on this, the innovations of this paper include the simultaneous use of gas and electricity energy network along with distributed generation sources on the standard network, simulation with real and variable values of loads and energy prices on the model MG, optimal dispatch of energy in the standard MG, for 24-hour, in terms of cost minimization and a practical look at the problem of energy management in MGs.

Based on this, in the second part of this article, the concept of an energy hub will be discussed. Then, in the third part, the modelling and expression of the relationships related to this research will be discussed, and based on that, the implementation procedure and related algorithms will be evaluated in detail. Next, in the fourth part of the article, the results obtained from this research will be explained. Based on the results of this discussion, the simultaneous minimization of the costs of electricity and gas MGs along with distributed generation sources and the production of each unit will be explained in detail. In the rest of the article, the obtained results will be discussed. At the end, the general conclusion will be expressed.

## 2. Energy hub concept

The Energy Hub (EH) concept was developed by a research team at the High Voltage Power Systems Laboratory in Zurich in the framework of a project called Vision of Future Energy Networks (VOFEN). This project aims to create a picture of future energy systems in the long term (20 to 30 years) using the Greenfield approach [1]. The main points of the VOFEN project can be summarized as follows:

- Moving towards multi-energy systems to benefit from the synergistic benefits of different energy carriers.

- Moving towards non-hierarchical structures.

- Moving towards integrated and connected energy systems

- In this paper, two concepts are presented to reach the main points mentioned.

- Energy connectors: combined transport of different energy carriers over longer distances in single transmission devices.

- Energy hub: conversion and storage of energy carriers in an integrated unit

Therefore, the concept of EH was introduced as a result of the VOFEN project and as an interface between consumers, producers, storage devices and transmission devices in different ways directly or through conversion equipment or multiple carriers.

Fig 1 shows the matrix model of the concept of EH to communicate different energy carriers in the input and output through the pairing matrix. Each element of the matrix represents the internal characteristics of EH, including the coefficients of connection and conversion of internal components of EH.

In further studies, the energy hub is defined as follows [1]:

- An EH is considered a unit in which several energy carriers can be converted, conditioned and stored.

- A more precise definition of EH can be stated as follows:

- EH is a unit that provides functions of input, output, conversion and storage of multiple energy carriers.

Sometimes, next to the energy hub, the word hybrid is used, which refers to the interaction of different energy carriers in EH [1].

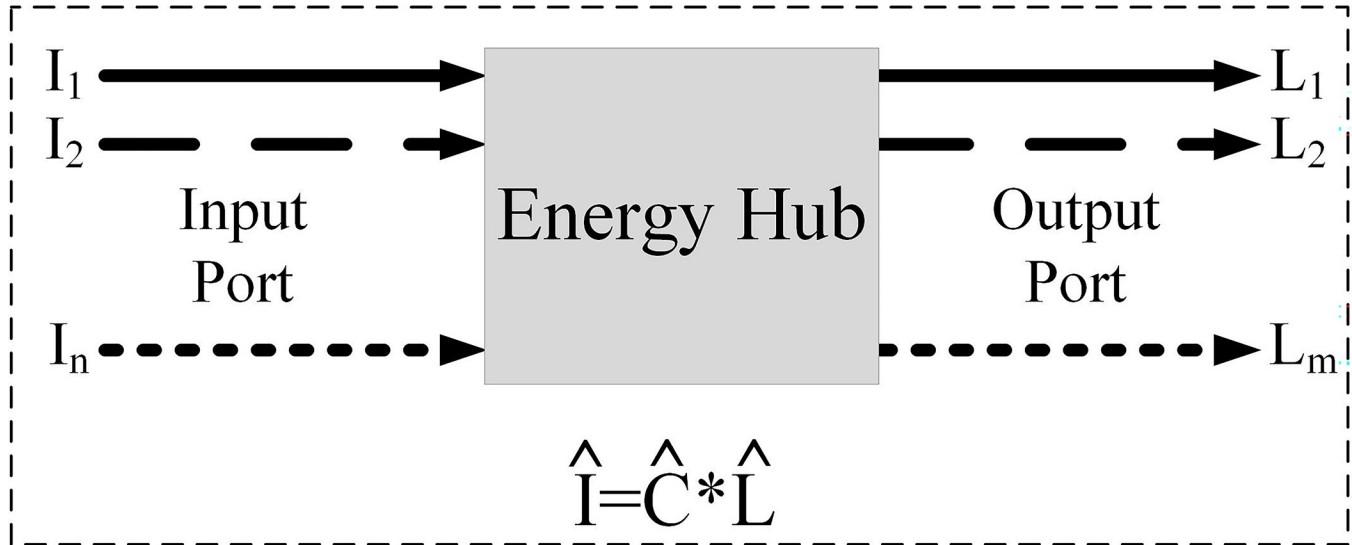

**Fig 1. Power conversion modeling through an energy hub.**

## 3. Implementation method

In this section, the modeling of the article's proposed procedure and the implementation method will be explained in detail.

### 3.1. Micro-grid modeling

In this part, the investigated MG models will be examined in detail along with the basic relationships of each part.

**3.1.1. Price modeling.** MGs participating in the market provide an hourly bid ($P_{iM,t}^{Bid}$) for the next day of energy trading. The MG is in consumer-state when the offered offer is negative, otherwise it will be producer. Changes between production and demand will lead to a power imbalance between the actual power and the proposed power ($\Delta P_t^{Imba}$) as shown in the following equation:

$$\Delta P_t^{Imba} = P_t^{Act} - \sum_{iM=1}^{M} P_{iM,t}^{Bid} \quad \forall t \tag{1}$$

So that $P_t^{Act}$ is the real power and it is obtained as the following equation:

$$P_t^{Act} = \sum_{iM=1}^{M} (P_{iM,t}^{Wind} + P_{iM,t}^{PV} + P_{iM,t}^{DG} + P_{iM,t}^{BESS}) + \sum_{j=1}^{L} P_{jt}^{IL} - \sum_{j=1}^{L} P_{jt}^{De} \tag{2}$$

where $P_{iM,t}^{Wind}$, $P_{iM,t}^{PV}$, $P_{iM,t}^{DG}$ and $P_{iM,t}^{BESS}$ are respectively the outputs of a wind turbine, solar panel, DG unit and battery energy storage systems (BESS) at time t and MG iM, $P_{jt}^{De}$ and $P_{jt}^{IL}$ show the power demand and Interrupted Load (IL) at time t and bus j, respectively. In addition, M and L are the total number of MGs and loads in the system. According to the amount of power deviation, the $Cost_t^{Imba}$ imbalance cost can be obtained as follows:

$$Cost_t^{Imba} = \begin{cases} \rho_t^{Re+}.\Delta P_t^{Imba}, & \Delta P_t^{Imba} < 0 \\ \rho_t^{Re-}.\Delta P_t^{Imba}, & \Delta P_t^{Imba} \geq 0 \end{cases} \tag{3}$$

where $\rho_t^{Re-}$ and $\rho_t^{Re+}$ provide the regulatory prices for the sale and purchase of electric energy in the RTB (Real-time balancing) market at time t, respectively. In addition, the DA market price is applied as a multiplier to calculate regulatory prices:

$$\begin{cases} \rho_t^{Re+} = (1 + \tau^+).\rho_t^{DA} \\ \rho_t^{Re-} = (1 + \tau^-).\rho_t^{DA} \end{cases} \tag{4}$$

where $\tau^-$ and $\tau^+$ are the relative differences between the sale and purchase price of electricity and the DA market price. The connection of electricity and gas networks along with electric and gas devices is shown in Fig 2. According to this figure, it can be seen that two renewable sources, wind and solar, are located in the grid together with the distributed gas generation source as sources of electric power generation. The gas distribution source is also connected to the gas network, and based on that, it is connected to the country's gas network, and it is necessary to have gas lines next to the electric lines. Therefore, according to this figure, it can be concluded that considering these two networks at the same time is very important.

**3.1.2. Gas system modeling.** In this part, gas system models and related relationships will be discussed.

*3.1.2.1. Line-pack modeling.* In the pipelines of gas networks, compressed gas energy is called Line-pack (LP) and has a direct relationship with the average pressure in the pipelines.

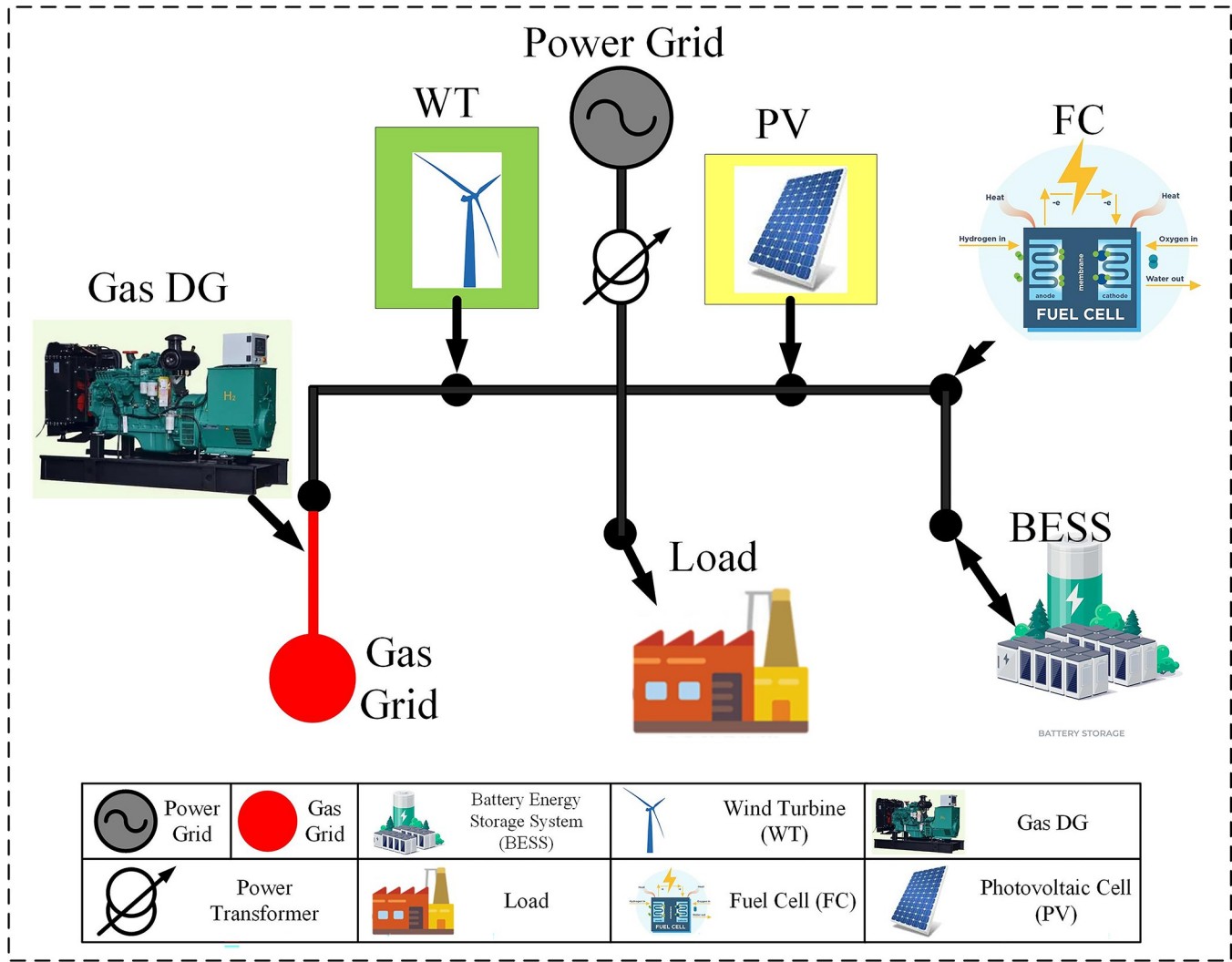

**Fig 2. Schematic of MGs in coupled gas and electricity networks.**

According to Boyle's law, gas in pipelines can be modeled as Eqs (A-1) and (A-2) in S1 Appendix, where Pr, $\zeta$, $v_{ij}$ and Te are gas pressure (kPa), density (Kg/m$^3$), volume (m$^3$) and temperature (K), respectively. Also, the values of $Pr_0$, $\zeta_0$, $v_{ij0}$ and $Te_0$ are pressure, density, volume and temperature in normal gas conditions, respectively. $Q^{Gas}$ and $Y^{Gas}$ values are gas compressibility coefficient and gas constant, respectively. $le_{ij}$ and $d_{ij}$ are the length and inner diameter of pipe i-j.

In dynamic conditions, the value of LP depends on the initial value of LP and the difference between gas consumed and supplied in the pipeline ($\Delta P_{ij,t}^{Gas}$). Gas consumers include non-electric gas loads, compressor loads, gas storage, DG gas loads. The gas network and storage are also considered as gas suppliers. In steady state, the size of LP is calculated as Eqs (A-3) and (A-4) in S1 Appendix, where $\theta_{ij}^{Initial}$ and $\theta_{ij,t}$ are the initial LP and the value of LP at time t, respectively. $\gamma$ is the gas volume to energy converter coefficient in normal conditions (GJ/m$^3$). $\Delta t$ is the time it takes to convert power into energy. $P_{Gas,ijt}^{De}$, $P_{Gas,ijt}^{Gen}$ and $P_{Gas,ijt}^{Comp}$ are demand, supplied gas and gas consumed by the compressor, respectively.

*3.1.2.2. Gas and compressor flow modeling.* Compressor and pipelines are two key elements of transmission system in gas networks. At steady state, the gas flow ($f_{ij}^{Gas}$) and direction

(Sgn$_{ij}$) are usually modeled by the Weymouth equation and depend on the gas pressure between the nodes. The flow equation of the pipeline between nodes i and j is presented as Eqs (A-5) and (A-6) in S1 Appendix, where P$_i$ (P$_j$) is the gas pressure at node i (j). Gr, χ$_{ij}$ and ζ are the gravity ratio, friction coefficient and air constant, respectively.

For a compressor, the gas flow (f$_{Gas,ij}$$^{Comp}$) and the amount of energy consumption are controlled by Eqs (A-7) and (A-8) in S1 Appendix, where P$_{Gas,ijt}$$^{Comp}$ and HP$_{Gas,ijt}$$^{Comp}$ are gas consumption and horsepower of a compressor, respectively. l$_{1, ij}$, l$_{2, ij}$ and l$_{3, ij}$ are the gas compressor coefficients between node i-j. Also, α$_{1, ij}$, α$_{2,ij}$ and α$_{3,ij}$ are coefficients related to compressor horsepower between i-j.

In addition, the limit of the compressor ratio PR$_{ijt}$$^{Comp,up}$ is as Eq (A-9) in S1 Appendix.

Eqs (A-5), (A-7) and (A-8) in S1 Appendix are the first group of non-linear equations in this paper, which should be converted into a suitable linear form for linear programs of integers. For (A-5) and (A-7), we apply the linearization process used with two and three variables, which is based on the Taylor series. In addition, for equations with the form of quadratic functions such as (A-8), the piecewise linear approximation method is used to convert the non-convexity of (A-8) into convexity.

*3.1.2.3. Distributional robust chance constrained (DRCC) modeling.* The DRCC method has been introduced as one of the powerful techniques for modeling probability-based constraint problems. Due to the fluctuating behavior of uncertain parameters, satisfying the constraints including them will also be possible. The DRCC method is used for problems with probabilistic constraints to not only deal with the uncertain nature of these problems, but also to maximize the MG profit with a certain probability according to the robustness of the system. The mathematical formulas of this method will be explained in this section. At first, it is assumed that the vector has random variables φ with the Eq (A-10) in S1 Appendix, where φ and σ$^2$ represent the mean and variance, respectively.

Family D is all distributions with mean φ and variance σ$^2$. We denote this family by D(φ,σ$^2$) and the DRCC method for each φϵ(0,1) according to the mentioned information is defined as Eq (A-11) in S1 Appendix.

This equation is equivalent to the adverb of the convex second order cone as Eq (A-12) in S1 Appendix where, φ is the probability of satisfying constraints with uncertain parameters.

## 3.2. Micro-grid optimum planning

In this section, optimal planning for MG will be described.

**3.2.1. Upper level (UL) problem.** In this paper, the market structure of day-ahead (DA) and real-time balance (RTB) is used. In the DA and wholesale gas markets, MGs maximize their profits by optimizing the bidding strategy within the time frame. The optimal MG planning objective function in the high-level problem (first stage) is formulated as follows:

$$
\begin{aligned}
OBJ_{iM}^{UP} = &\sum_{t=1}^{T} \rho_t^{DA}.P_{iM,t}^{Bid}.\Delta t + \\
&+[(\sum_{j=1}^{L}\sum_{t=1}^{T} \rho_t^{RT}.P_{jt}^{De}.\Delta t) - (\sum_{j=1}^{L}\sum_{t=1}^{T} \rho_t^{RT}.P_{jt}^{IL}.\Delta t)] + \\
&+\sum_{t=1}^{T} Cost_t^{Imba} - \sum_{j=1}^{L}\sum_{t=1}^{T} Cost_{jt}^{IL} - \sum_{t=1}^{T} Cost_{iM,t}^{BESS} - \sum_{t=1}^{T} Cost_{iM,t}^{DG} + \\
&+\sum_{i=1}^{L}\sum_{j=1}^{L}\sum_{t=1}^{T} \rho_t^{Gas,retail}.P_{Gas,ijt}^{De} - \sum_{i=1}^{L}\sum_{j=1}^{L}\sum_{t=1}^{T} \rho_t^{Gas}.P_{Gas,ijt}^{Gen} \quad \forall iM
\end{aligned}
\tag{5}
$$

 

where $\rho_t^{RT}$ and $\rho_t^{Gas,retail}$ are the retail prices of electricity and gas, and $\rho_t^{Gas}$ is the price of gas in the wholesale gas market. T is the timing interval of DA. $P_{Gas,ijt}^{De}$ and $P_{Gas,ijt}^{Gen}$ are the amount of gas demand and production. In the objective function, the first term is the income (cost) of selling (buying) electric energy from (by) the MG to (from) the distribution network. The second term is the income from the sale of electrical energy to consumers. The next four terms are the imbalance cost, IL, BESS, and DG units, respectively. The last two periods are the income from selling gas to gas consumers and the cost of buying gas from the wholesale gas market.

According to what was said, when $\Delta P_t^{Imba}<0$, the MG needs to purchase energy shortages in the RTB market, and the sign of $Cost_t^{Imba}$ in (3) will be negative, which is the imbalance cost for MGs to apply the mentioned cost in the objective function. The sign of the third part (5) must be positive, which becomes a negative sign in the presence of $Cost_t^{Imba}$. After solving the problem, the value of the mentioned variables is determined and all of them are considered to minimize the power deviation cost in the second stage problem. In the high-level DA scheduling problem, the model is considered as a here-and-now decision process.

The cost function $Cost_{jt}^{IL}$ has a direct dependence on the amount of interrupted load and is modeled as Eq (A-13) in S1 Appendix, where $\alpha_{1j}^{IL}$ and $\alpha_{2j}^{IL}$ are coefficients based on IL cost.

The cost function of $Cost_{iM,t}^{BESS}$ is considered as a linear function and can be modeled as Eqs (A-14) and (A-15) in S1 Appendix, where $P_{iM,t}^{BESS}$ is the charge or discharge of power in the BESS at time t in the iM MG. $Es_{iM,t}^{BESS}$ and $\eta^{LIf}$ are stored energy and leakage loss coefficient in BESS, respectively. $BLD_{iM}^{BESS}$ is the BESS lifetime degradation cost factor. $LCN^{BESS}$, $BIC$ and $E_{RC}^{BESS}$ are the life cycle number, investment cost and rated energy capacity of BESS, respectively. The first and second terms of (A-14) derive the cost of the BESS from its charge/discharge and the energy stored in the BESS, taking into account the leakage loss factor.

The DG unit is considered as a gas burning system and its cost function is formulated by a quadratic function as Eqs (A-16) and (A-17) in S1 Appendix, where $X_{iM,t}^{DG}$, $X_{iM,t}^{SD}$ and $X_{iM,t}^{SU}$, respectively provide binary variables that provide DG on/off status, shutdown and startup decisions. $SUC_{iM,t}^{DG}$ and $SDC_{iM,t}^{DG}$ are the startup and shutdown costs of DG units. $\alpha_{1j}^{DG}$, $\alpha_{2j}^{DG}$ and $\alpha_{3j}^{DG}$ are the coefficients based on the unit cost of DG in iM MG at time t.

To obtain the uncertainty parameters, the DRCC method should be applied to the problem formulations. At first, the total profit of the MG F in the DA scheduling problem is given as Eq (A-18) in S1 Appendix.

To use the DRCC method, the auxiliary variable Z to maximize the profit of the MG is defined as Eq (A-19) in S1 Appendix, where $\varphi$ is the level of confidence determined by the decision maker based on the strength of the system. Therefore, the main objective function after applying DRCC is as Eq (A-20) in S1 Appendix, where $\sigma\rho_t^{DA}$ and $\sigma\rho_t^{Gas}$ are the standard deviations of DA market and gas prices, respectively.

Obtaining a certain amount of profit for MGs is guaranteed by the DRCC method; because in the worst case, uncertain parameters in the system are obtained by considering specific values for the mean and variance of each uncertain parameter and a certain value from it. The confidence level determined by the decision maker. If the mean and variance values for uncertain parameters change based on the energy market and geographic conditions, or the level of confidence changes by the decision maker, it is obvious that the benefit will be for MGs. In new conditions, the system is guaranteed due to considering the worst cases of all uncertain parameters. The complete constraints related to the high-level problem are also stated as follows.

*3.2.1.1. Electric energy balance limitation*. The limit of electrical energy balance is expressed as Eq (A-21) in S1 Appendix.

*3.2.1.2. Wind power limitation.* The limit of wind power is expressed as Eqs (A-22) and (A-23) in S1 Appendix, where, $Q_{iM,t}^{Wind}$ is the output reactive power of the wind turbine at bus i and at time t.

After applying the DRCC method, the new wind power limit will be as Eq (A-24) in S1 Appendix, where $P_{iM,t}^{Wind,UP}$, $\omega$ and $\sigma P_{iM,t}^{Wind,UP}$ are the upper limit, confidence level and standard deviation of wind power, respectively.

*3.2.1.3. Solar cell power limitation.* The limitation of solar cell power is expressed as Eq (A-25) in S1 Appendix, where SOL, $S_{PV}$, and $\eta_{PV}$ are solar radiation, PV panel size, and PV efficiency, respectively.

After applying the DRCC method, the limitation of the new PV panel is expressed as Eq (A-26) in S1 Appendix, where ꚃ and $\sigma_{SOLt}$ are the confidence level of PV and the standard deviation of $SOL_t$ at time t, respectively.

*3.2.1.4. Dispersed production unit limitation.* The limitation of distributed production unit is stated as Eqs (A-27)-(A-37) in S1 Appendix, where PiM, $P_{iM,t}^{DG}$ and $P_{iM,t}^{DG}$ show, respectively, the lower and upper limit of DG power in iM MG at time t. $P_{Gas,iM,t}^{De,DG}$ is the gas demand of the DG unit. $HRa^{Gas}$ is the gas heating rate (MW/GJ). $Ra_{iM}^{DG,Up}$ and $Ra_{iM}^{DG,Down}$ are the up and down ramps of DG in iM MG. $MUT_{iM}$ and $MDT_{iM}$ are minimum upper and lower limits of DG. e is an index to model the minimum upper and lower bounds running from 1 to Max{$MUT_{iM}$,$MUT_{iM}$}.

*3.2.1.5. BESS energy storage system limitation.* The limitation of the energy storage system is stated as Eqs (A-38)-(A-42) in S1 Appendix, where, $\eta^{Clf}$ is the loss coefficient related to the charging or discharging state of the BESS. In this research, the value of the loss coefficient is assumed to be constant. $P_{iM,t}^{Chr,BESS}$ and $P_{iM,t}^{Dis,BESS}$ are the maximum charge and discharge power of BESS. a and b are auxiliary variables that are used to remove the absolute function. Eq (A-38) in S1 Appendix shows the limit of BESS energy balance. Eqs (A-39) to (A-40) in S1 Appendix show the state of load restrictions and (A-41) shows the discharge or charging capacity. Finally, the final and initial energy stored in the BESS are given in (A-42).

*3.2.1.6. IL flow limitation.* To ensure the stability of the power grid in emergency situations (for example, when the sum of electricity produced and energy received from the main grid is less than energy consumption), the possibility of load reduction is also considered for energy balance in the grid. The presence of random producers whose restrictions are as Eqs (A-43)-(A-44) in S1 Appendix. Eqs (A-43) and (A-44) in S1 Appendix state that the value of IL must be kept within the permissible range.

*3.2.1.7. Electric energy network limitation.* The eqations of AC current in each bus and at time t are formulated as Eqs (A-45)-(A-51) in S1 Appendix, where $P_{it}^{Injecte}(V_{it},\theta_{it})$ and $Q_{it}^{Injecte}(V_{it},\theta_{it})$ correspond to active and reactive power injection in bus i at time t, respectively. PitDe and QitDe show active and reactive power demand. Eqs (A-49) to (A-51) in S1 Appendix express the allowed changes of mixed power, voltage magnitude and voltage angel, respectively.

*3.2.1.8. Gas network limitation.* The limitation of the gas network is expressed as Eqs (A-52)-(A-55) in S1 Appendix, where $P_{Gas,ijt}^{De}$ and $P_{Gas,ijt}^{De,Heat}$ are the total energy demand and heating energy load at time t, respectively. Eqs (A-52) and (A-53) in S1 Appendix express gas node balance and demand limits, respectively. Eqs (A-54) and (A-55) in S1 Appendix express LP changes, initial and final energy of LP limits, respectively.

*3.2.1.9. Connection exchanges limitation.* The limit of connection exchanges is stated as Eq (A-56) in S1 Appendix, where, the lower and upper lines are the minimum and maximum amount of energy exchange in the energy market, respectively.

**3.2.2. Lower level (LL) problem.** In the RTB market, the power balance can be created by optimal planning in the timing horizon. It is assumed that this market is very high. Every 5 minutes after determining the value of binary variables by solving the proposed optimal model

in the high-level problem, all MGs should adjust their controllable devices such as DG unit to compensate for power deviations in the real-time balancing market. In the lower level problem, the relationship is expressed as follows:

$$Min \ OBJ^{LO} = \sum_{t=1}^{T} Cost_t^{Imba} \qquad (6)$$

where $Cost_t^{Imba}$ is the cost of disequilibrium in the lower level problem. The complete constraints of Eq (6) include Eqs (A-21) to (A-30) in S1 Appendix, Eqs (A-38) to (A-41) in S1 Appendix and Eqs (A-43) to (A-55) in S1 Appendix.

Since the main goal in the second stage is to minimize the imbalance costs by minimizing the amount of power deviations, the value of $P_{iM,t}^{Bid}$ (which is determined in the problem of the first stage) in Eq (A-57) in S1 Appendix is used for this purpose. Therefore, the value of $P_t^{RT}$ is updated by adjusting the DERs schedules in (A-58) considering the value of $P_{iM,t}^{Bid}$ with the aim of minimizing the power deviations without changing the state of the DG units that were previously determined in the first. Phase problem by minimizing the amount of power deviations, the amount of unbalance costs is also minimized according to (A-59).

### 3.3. Micro-grid energy price

The average of energy price (EP) is shown in Fig 3. This value is adopted in the year according to the data obtained from 2020. As can be seen, the value of this index is changing during one

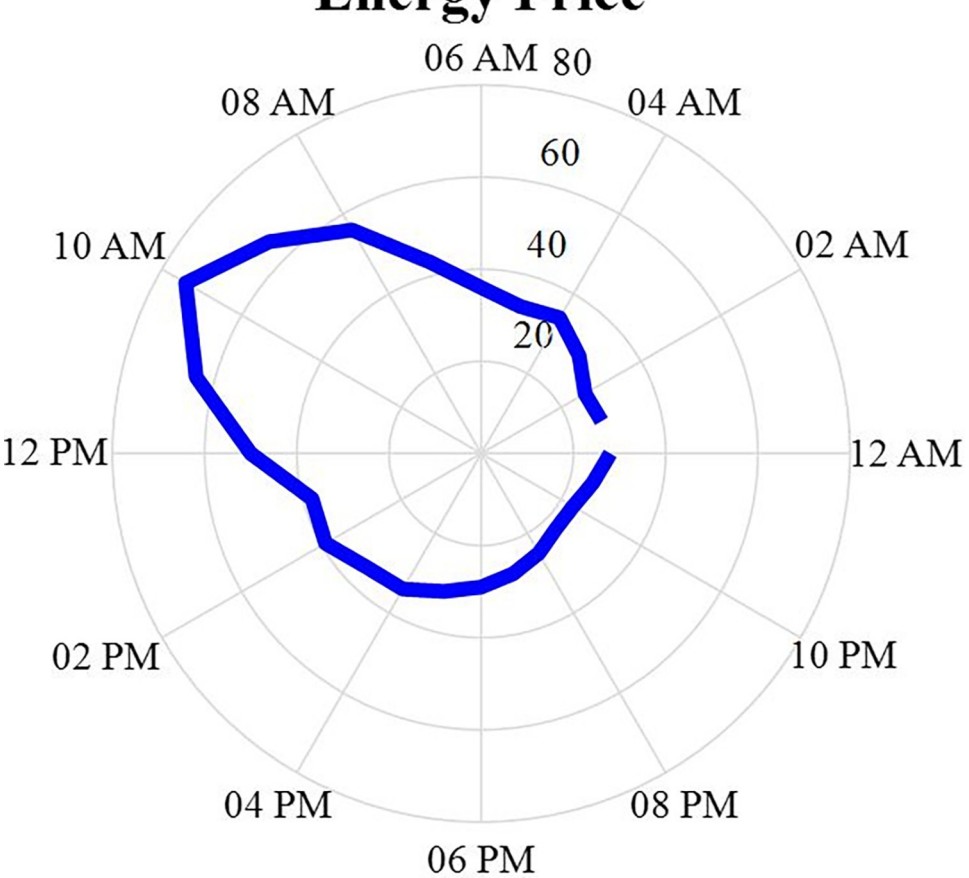

**Fig 3. Energy price during 24 hours.**

day, and during peak hours, the EP consumption is higher than during off-peak hours. These values have an impact on the cost of the MG. In other words, if the load of the MG is high during peak hours, the cost of the MG will increase.

Now, by achieving the production capacity of each power plant according to the above procedure and also considering the EP according to the above figure, the production cost of each power plant during one day can be obtained according to the following relationship:

$$Price^j = \sum_{i=1}^{24} P_i^j EP_i, \quad \begin{matrix} i = hour \\ j = Power\ Plant\ Type \end{matrix} \tag{7}$$

where $Price^j$ is the cost of the j-th power plant, $P_i^j$ is the power value of the same power plant in the i-th hour, and $EP_i$ is the energy price in the i-th hour.

## 4. Simulation results

The simulation related to this research is coded in Games software and as a mixed integer non-linear programming (MINLP). The reason for using this software is its ability to solve non-linear problems and optimization. The efficiency of gas turbines and fuel cells (FC) depends on their working point, and considering the exact model of these sources and the relationships related to the calculation of their fuel consumption is non-linear. On the other hand, a binary variable has been used to show the charging and discharging state of the storage tank and the on-and-off state of the gas turbines. Therefore, our problem is an MILP. There are several solvers in Games to solve MILP problems:

- DICOPT solver has a very good speed.

- The COUENNE solver has been used to solve the problem, which has a good performance, but the speed of its solution is a little slow.

According to what was said, to electrify the load in this article, it is necessary to feed this power by the following power plants:

- Production power of micro-turbine 1 (MT1)

- Production power of micro-turbine 2 (MT2)

- Production power of micro-turbine 3 (MT3)

- Production power of solar cell

- Production power of wind turbine

- Production power of fuel cell

- Power of charging and discharging energy storage

The simulation performed in this paper was tested on a 33-bus IEEE radial distribution system and its information can be accessed in [32]. The general schematic of this micro-grid is according to Fig (2). Also, the load profile of this network has been used based on [33].

The safety pressure and diameter for gas pipelines are 140 kPa and 660 mm, respectively. Information related to wind turbine sources as well as distributed generation units is given in [34]. In addition, the price of natural gas can be seen in [35].

Now, the results of this simulation will be examined in Gomes software, including the amount of load power and each of the above production powers.

## 4.1. Power consumer values

The amount of consumer power or consumption demand, in the network under investigation, during 24 hours a day, is specified in this section. This consumer data can be checked in the following two scenarios:

- Optimized demand

- Un-optimized demand

These load data during the period of 24 hours and in the two stated scenarios will be in the form of Fig 4. According to this figure, it can be seen that the non-optimized load has more load power than the optimized load power. Also, the load capacity increases after 16:00 and the amount of load in state un-optimized demand shows less fluctuation in other words, consumption optimization has not been done. In the morning hours, the amount of load power is high.

## 4.2. Power producer values

As mentioned, the data values of the producers include 7 production sources of MT1, MT2, MT3, solar cell, wind turbine, fuel cell and storage energy source. Based on this, in this section and in the following, the details of the production power of each of these sources will be discussed after optimization in GAMS software.

**4.2.1. MT1 power producer values.** The production power value of MT1 after optimization is specified in this section in the form of a correct mixed non-linear programming in

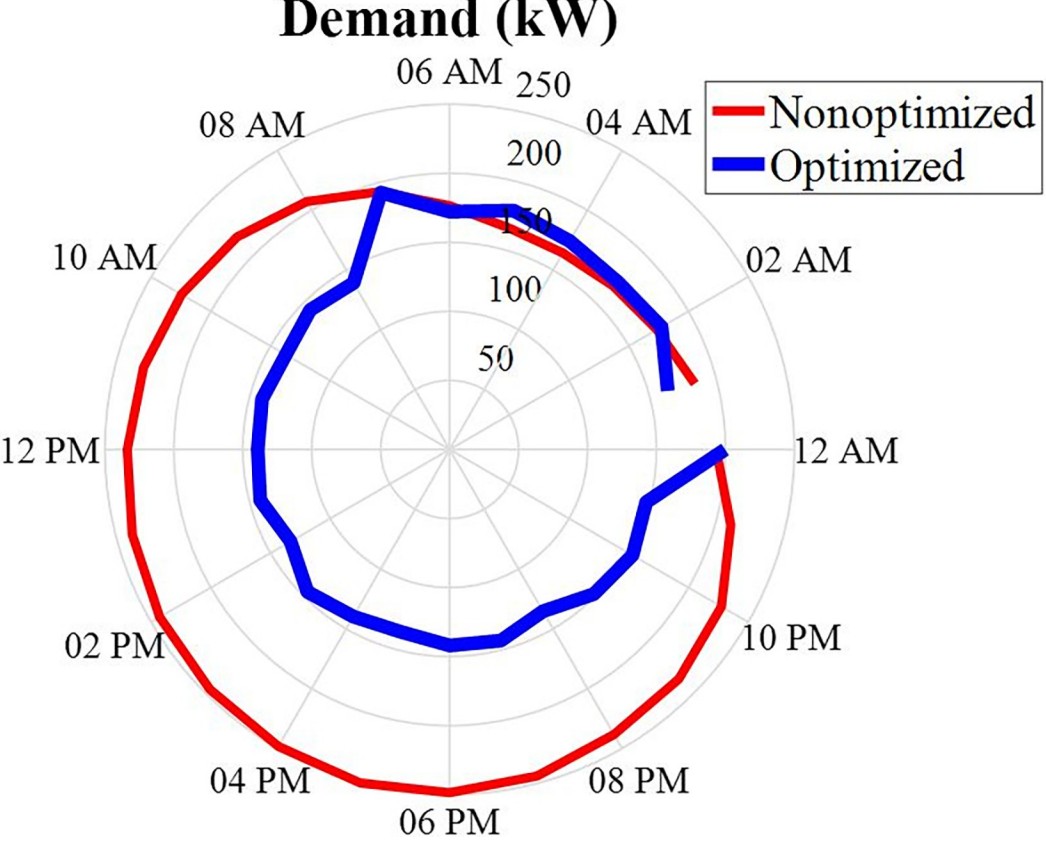

**Fig 4. Load data during 24 hours.**

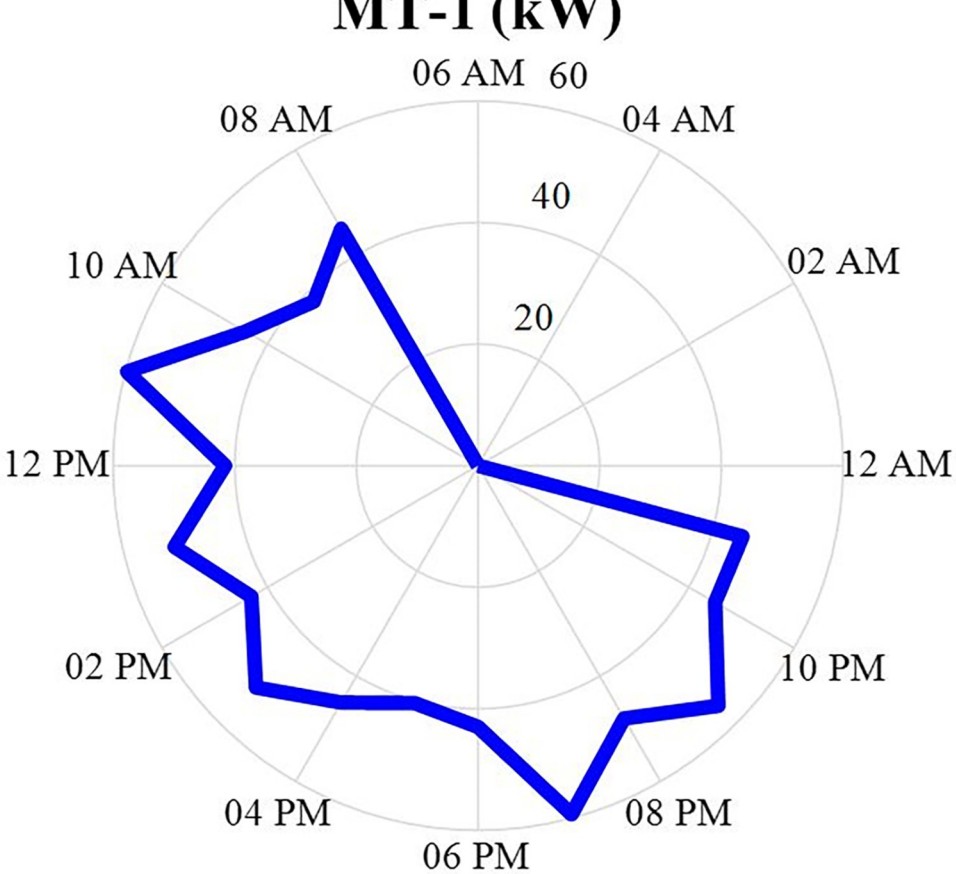

**Fig 5. MT1 production power data during 24 hours.**

GAMS software and during 24 hours. This production data will be in the Fig 5. According to this figure, it can be seen that this production source produces more power in the morning until the end of the day and turns off at night. The reason for the high fluctuation in this power generation is the peak load during the day and the related optimization issue.

**4.2.2. MT2 power producer values.** The production power value of MT2 after optimization is specified in this section in the form of a correct mixed nonlinear programming in GAMS software and during 24 hours. This production data will be in the Fig 6. According to this figure, it can be seen that this production source produces more power in the morning until the end of the day and turns off at night.

**4.2.3. MT3 power producer values.** The production power value of MT3 after optimization is specified in this section in the form of a MINLP in GAMS software and during 24 hours a day. This production data will be in the Fig 7. According to this figure, it can be seen that this production source produces more power in the morning until the end of the day and turns off at night. According to Figs 5–7, it can be seen that three situations MT1-MT3 have similar behavior and all three have little production in the early hours of the day and their production increases in the later hours.

**4.2.4. Photovoltaic power producer values.** The amount of solar cell production power after optimization is specified in this section in the form of a correct mixed nonlinear programming in the Gomes software and during 24 hours a day. This production data will be in the Fig 8. According to this figure, it can be seen that this production source starts producing

# MT-2 (kW)

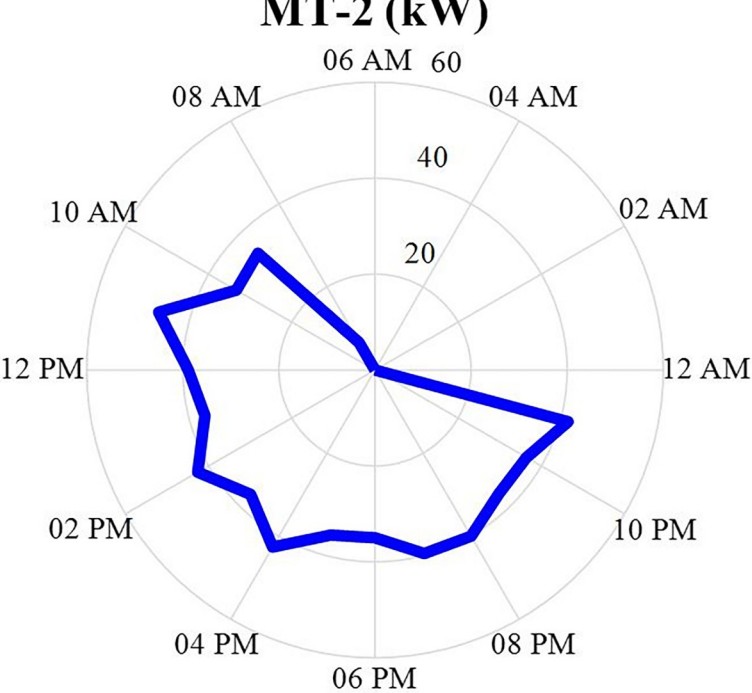

**Fig 6. MT2 production power data during 24 hours.**

# MT-3 (kW)

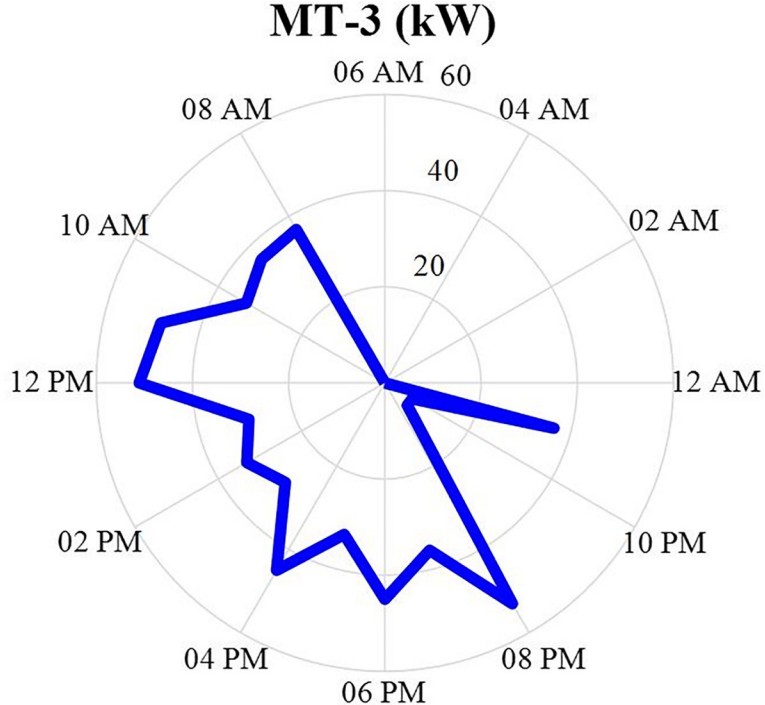

**Fig 7. MT3 production power data during 24 hours.**

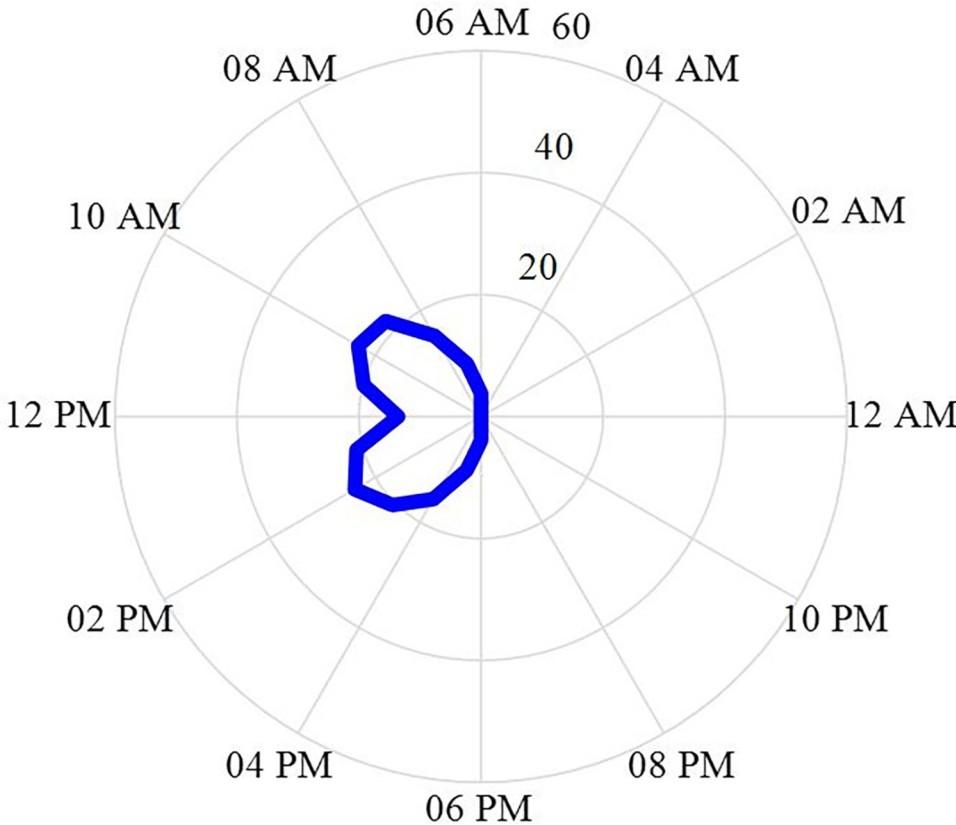

**Fig 8. PV production power data during 24 hours.**

power during the hours of the day when the sun starts to shine, and when the sun's radiation decreases, its production power becomes zero.

**4.2.5. Wind turbine power producer values.** The amount of wind turbine production power after optimization is specified in this section in the form of a correct MINLP in Gomes software and during 24 hours a day. This production data will be in the Fig 9. According to this figure, it can be seen that in some hours of the day and night when the wind speed increases, the amount of power produced from this source has also increased, and this amount of power may be completely different on another day. In other words, the power produced by the wind turbine source depends on the wind speed and does not depend on the hours of the day.

**4.2.6. Fuel cell power producer values.** The output power of the fuel cell after optimization is specified in this section in the form of a MINLP in GAMS software and during 24 hours a day. This production data will be in the Fig 10. According to this figure, it can be seen that the power generated from this source of power generation is turned off at night and does not generate power.

**4.2.7. Energy storage power producer values.** The amount of power and energy charged and discharged of the energy storage after optimization is specified in this section in the form of a correct MINLP in GAMS software 24 hours a day. The data of the charged and discharged power of the energy storage will be as shown in fig 11. Positive values in this figure indicate the discharge status of the energy storage system and negative values indicate the charging status

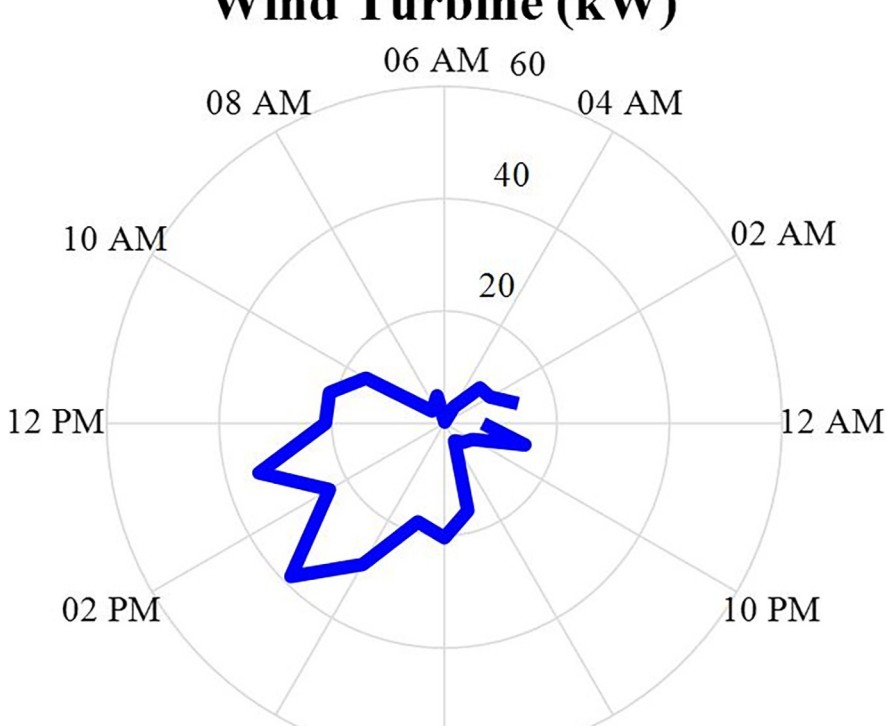

**Fig 9. Wind turbine production power data during 24 hours.**

of the energy storage system. According to this figure, it can be seen that the power charged or discharged from this energy source depends on the difference between the power produced from other sources and the power consumed by the load. In other words, if the amount of power generation is more than the load, the energy storage system will be charged, and on the contrary, if the amount of load is more than the power production, the energy storage system will be discharged. Also, the energy data of the energy storage will be in the Fig 12.

## 5. Discussion

In this section, the results obtained in the previous sections will be discussed. Based on this, at first, we will obtain the total production power from the sources of MT1, MT2, MT3, solar energy, wind energy, and fuel cells. Then, the difference between this value with the charge and discharge power values of the energy storage system is obtained to see whether this energy storage source correctly detects charging and discharging or not.

For this purpose, fig 13 shows the total production power from the 6 sources mentioned above during a period of 24 hours a day.

Fig 14 also shows the difference in the amount of power produced by six power generation and the load in the non-optimized scenario. According to this figure, it can be seen that this amount of power expresses the amount of charging and discharging of the energy storage source. In other words, this difference is the lack or excess of power in the network, which

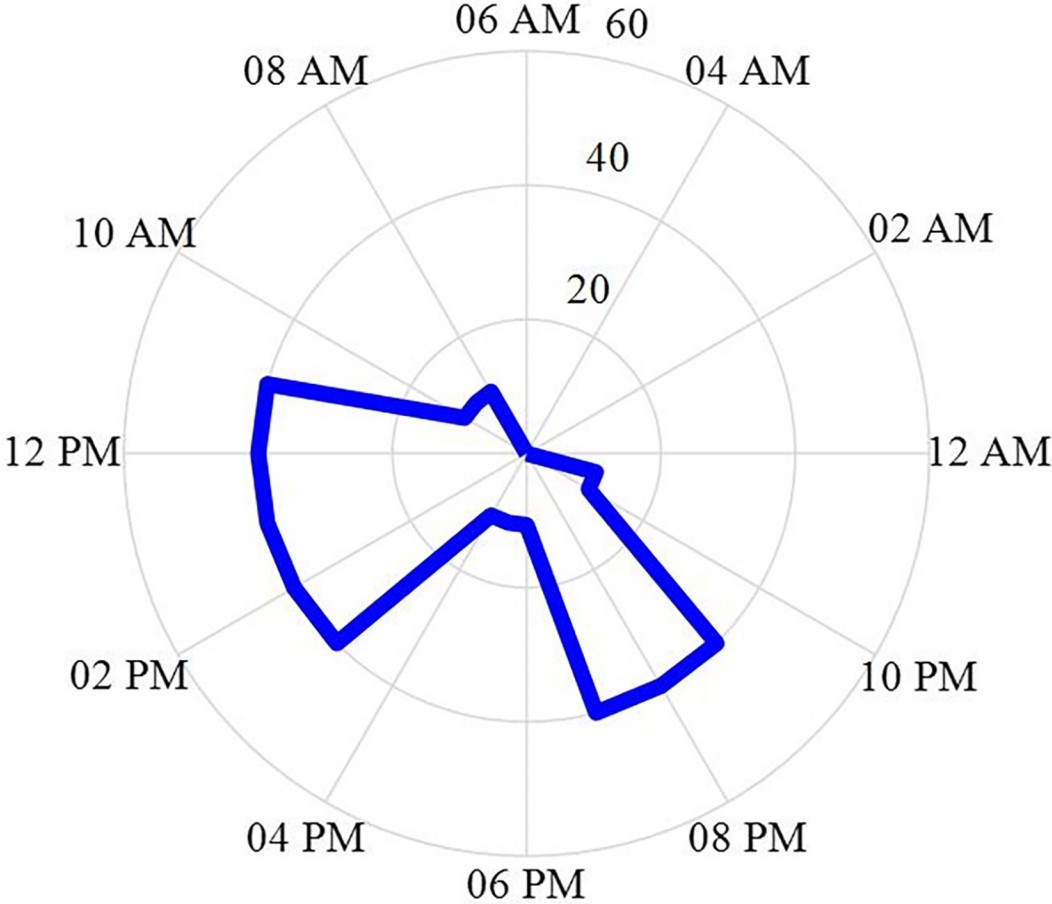

**Fig 10. Fuel cell production power data during 24 hours.**

needs to be charged and discharged by the energy storage source. According to this figure, it can be seen that this figure is similar to Figs (5–9) with an acceptable approximation. The relative difference between these two forms is caused by the inertia of the energy storage source that will change its state.

Now, according to fig 14, it can be seen that there is no significant difference between the power produced by the six production sources and the load power, and this optimization procedure can be fully used.

Also, according to the reference [2] that the data was extracted according to this reference, it can be seen that the network costs will be significantly reduced with the optimization done in this article. In other words, if the appropriate wind and solar production power was not used optimally in this article, as in reference [2], it was necessary to use the energy of micro-turbines to produce electric power, which is extremely expensive. It increased the network.

The amount of produce price of a power plant in the MG is obtained according to Table 2. As can be seen:

- The economic efficiency of micro-turbines is higher than other power plants due to continuous power generation.

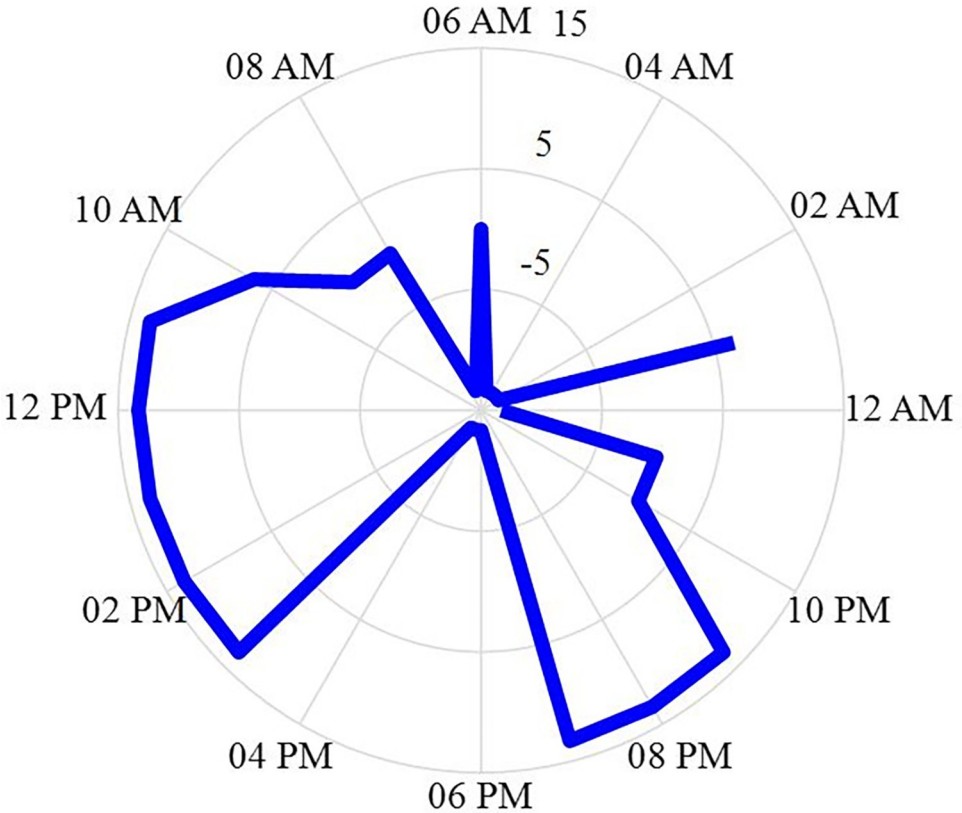

**Fig 11. Charging and discharging energy storage production power data during 24 hours.**

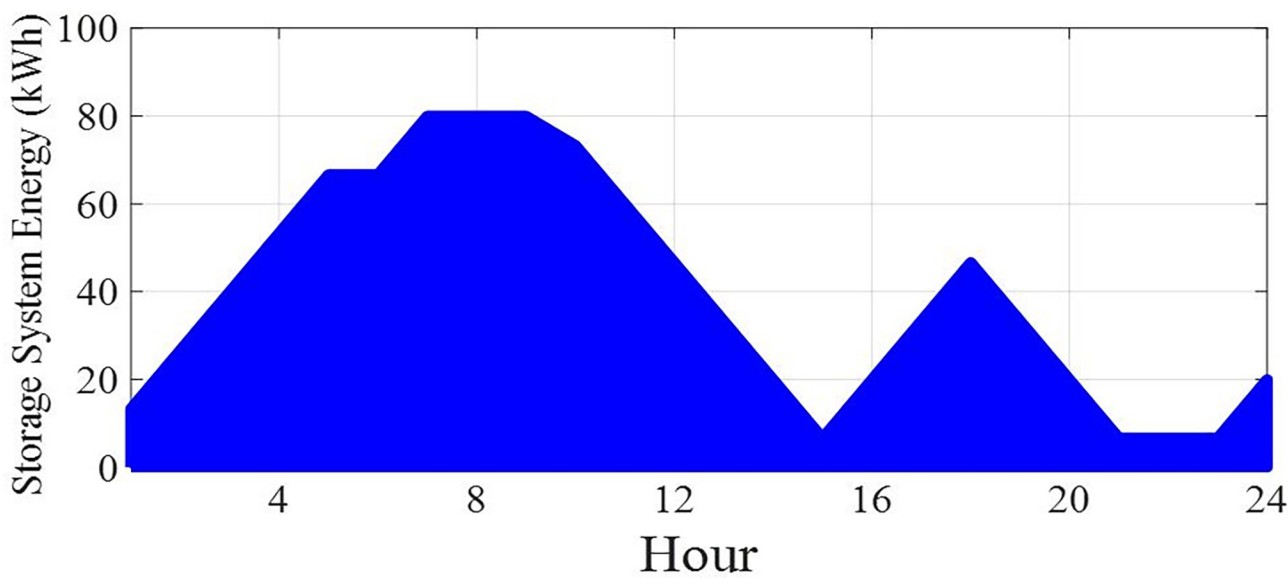

**Fig 12. Energy data of energy storage production during 24 hours.**

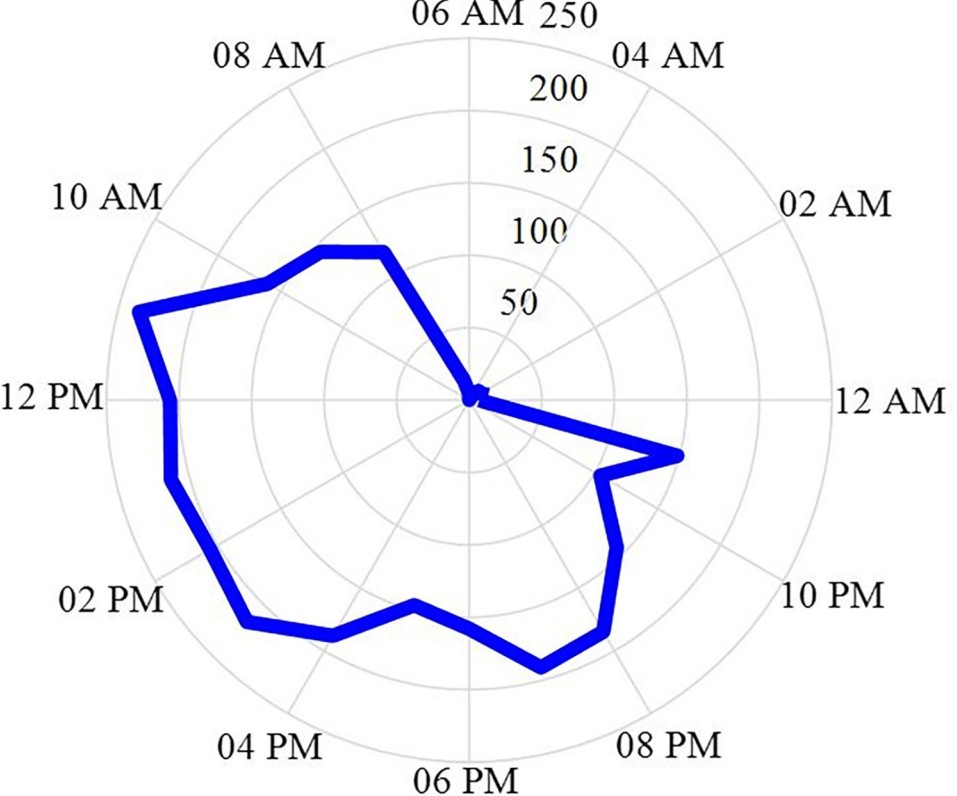

**Fig 13. Total production capacity during 24 hours.**

- The economic efficiency of the solar cell is relatively suitable because it produces power during peak load times (note that the solar cell produces power only in limited hours of the day and yet it has a significant amount of economic efficiency).

- The amount of power produced by the battery storage is low due to being discharged every moment, although it plays a significant role in smoothing the production power of the network.

**Table 2. *Price of power plant*.**

| No. | Power Plant Type | Price (USD) |
|---|---|---|
| 1 | MT1 | 29900 |
| 2 | MT2 | 22687 |
| 3 | MT3 | 23304 |
| 4 | PV | 9875.7 |
| 5 | WT | 12274 |
| 6 | FC | 15630 |
| 7 | BESS | 847.2 |

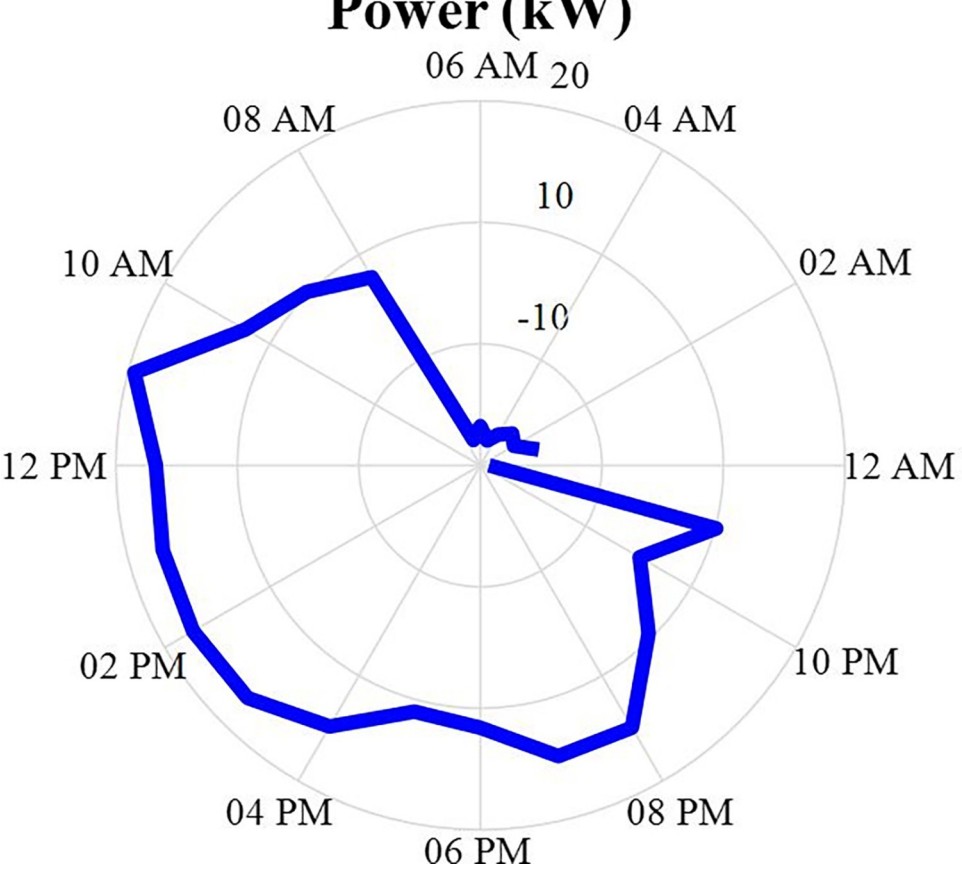

**Fig 14. All power production and load power differences in non-optimized scenario during 24 hours.**

## 6. Conclusion

In this article, an attempt was made to code the simulation in the form of an MILP using GAMS software. Based on this and according to the output of this software, the amount of power difference between generators and load using GAMS problem optimization, the charge and discharge values of energy storage have been expressed. In other words, this difference is the lack or excess of power in the network, which needs to be charged and discharged by the energy storage source. The results of this optimization were verified according to the charging and discharging output of the energy storage device. The relative difference was caused by the inertia of the energy storage source. By precisely defining the optimization problem and extracting the objective function, the amount of production of each electrical energy generating unit with two issues of electricity and gas during 24 hours was accurately obtained. According to the results of this article, it is possible to accurately determine the amount of production of each resource to reach the optimal cost amount during the 24 hours. With this analysis, network costs will be significantly reduced by combining gas and electricity. In other words, it is possible to go to gas power plants during the hours when electricity is needed and use relatively cheaper energy such as solar energy during the day, thus increasing network costs. With optimal energy consumption and management of the production side, network costs can be optimized and this cost can be transferred to investment in other sectors. The method proposed in this article is based on mathematics and due to the use of software based on optimization, the accuracy of its results is high.

## Supporting information

**S1 Data. Microgrid_Gas.gms. GAMS code of micro-grid.**
(M)

**S2 Data. Power_Tolid.Masraf.m. MATLAB code of figure.**
(GMS)

**S3 Data. Main.m. MATLAB code of figure.**
(M)

**S1 Appendix.**
(DOCX)

## Author Contributions

**Conceptualization:** Shahryar Behnia, Saeed Kharrati.

**Data curation:** Shahryar Behnia.

**Formal analysis:** Shahryar Behnia.

**Funding acquisition:** Shahryar Behnia.

**Investigation:** Shahryar Behnia.

**Methodology:** Shahryar Behnia.

**Project administration:** Saeed Kharrati.

**Software:** Shahryar Behnia.

**Supervision:** Saeed Kharrati, Farshad Khosravi, Abdollah Rastgou.

**Validation:** Shahryar Behnia.

**Visualization:** Shahryar Behnia, Farshad Khosravi, Abdollah Rastgou.

**Writing – original draft:** Shahryar Behnia.

**Writing – review & editing:** Shahryar Behnia.

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
