## [Decision Letter · Decision Letter 0]

16 Apr 2024

PONE-D-24-09657Simultaneous Costs Minimizing in Electricity and Gas Micro-grids with the Presence of Distributed GenerationPLOS ONE

Dear Dr. Kharrati,

Thank you for submitting your manuscript to PLOS ONE. After careful consideration, we feel that it has merit but does not fully meet PLOS ONE’s publication criteria as it currently stands. Therefore, we invite you to submit a revised version of the manuscript that addresses the points raised during the review process.

We look forward to receiving your revised manuscript.

Kind regards,

Dr. Mohit Bajaj

Academic Editor

PLOS ONE

Journal Requirements:

2. Please note that PLOS ONE has specific guidelines on code sharing for submissions in which author-generated code underpins the findings in the manuscript. In these cases, all author-generated code must be made available without restrictions upon publication of the work. 

Please review our guidelines at https://journals.plos.org/plosone/s/materials-and-software-sharing#loc-sharing-code and ensure that your code is shared in a way that follows best practice and facilitates reproducibility and reuse.

3. We note that your Data Availability Statement is currently as follows: 

"All relevant data are within the manuscript and its Supporting Information files."

6. We note that Figure 2 in your submission contain copyrighted images. All PLOS content is published under the Creative Commons Attribution License (CC BY 4.0), which means that the manuscript, images, and Supporting Information files will be freely available online, and any third party is permitted to access, download, copy, distribute, and use these materials in any way, even commercially, with proper attribution. For more information, see our copyright guidelines: http://journals.plos.org/plosone/s/licenses-and-copyright.

1) You may seek permission from the original copyright holder of Figure 2 to publish the content specifically under the CC BY 4.0 license. 

2) If you are unable to obtain permission from the original copyright holder to publish these figures under the CC BY 4.0 license or if the copyright holder’s requirements are incompatible with the CC BY 4.0 license, please either i) remove the figure or ii) supply a replacement figure that complies with the CC BY 4.0 license. Please check copyright information on all replacement figures and update the figure caption with source information. 

If applicable, please specify in the figure caption text when a figure is similar but not identical to the original image and is therefore for illustrative purposes only.

**Additional Editor Comments:**

The reviewers have raised the following concerns in regard to experimental design. Thus, we ask that you revise your manuscript along the lines they have suggested, in order to address this issue. 

Reviewers' comments:

Reviewer's Responses to Questions

**Comments to the Author**

1. Is the manuscript technically sound, and do the data support the conclusions?

Reviewer #1: Yes

Reviewer #2: No

2. Has the statistical analysis been performed appropriately and rigorously? 

Reviewer #1: No

Reviewer #2: I Don't Know

3. Have the authors made all data underlying the findings in their manuscript fully available?

Reviewer #1: No

Reviewer #2: No

4. Is the manuscript presented in an intelligible fashion and written in standard English?

Reviewer #1: Yes

Reviewer #2: No

5. Review Comments to the Author

Reviewer #1: 1- The abstract is very long, write this section briefly.

2- Compare your work with other works in a table in the article.

3- It is suggested to check the following studies in the introduction:

# https://doi.org/10.1002/er.5709

#https://doi.org/10.1007/s12652-020-02322-2

#https://doi.org/10.1155/2023/9042957

4- What is the main difference between your work and other works? In other words, specify the innovation of the article clearly in the article.

5- The language of the article needs to be revised.

6- Analyzing the results is not enough. In other words, the explanation section about the results is incomplete.

7- What test network is used in the article?

8- The data related to solar and wind were extracted from which study?

9- The conclusion does not support the simulation results.

Reviewer #2: 1- The manuscript needs substantial revision for language and grammar.

"scattered production"= Distributed generation

"This article is done on a multi-objective network"

"transformation of passive and passive"

" these two energies" these two energy carriers"

"mixed correct nonlinear programming" = mixed integer linear programming known as MILP.

"Gems"=GAMS

1- using appendix for equations makes the review of the paper hard.

3- overall the paper is not well organized.

6. PLOS authors have the option to publish the peer review history of their article (what does this mean?). If published, this will include your full peer review and any attached files.

Reviewer #1: No

Reviewer #2: No

---

## [Author Response · Author response to Decision Letter 0]

12 May 2024

Thank you so much for sending the reviewers comments related to manuscript entitled “Simultaneous Costs Minimizing in Electricity and Gas Micro-grids with the Presence of Distributed Generation”. We applied the reviewers’ comments to improve the quality of the paper. Responses to the reviewers’ comments are presented in the same order as the comments have been addressed. All answers are sent in the attached file.

---

## [Decision Letter · Decision Letter 1]

2 Aug 2024

PONE-D-24-09657R1Simultaneous Costs Minimizing in Electricity and Gas Micro-grids with the Presence of Distributed GenerationPLOS ONE

Dear Dr. Kharrati,

Thank you for submitting your manuscript to PLOS ONE. After careful consideration, we feel that it has merit but does not fully meet PLOS ONE’s publication criteria as it currently stands. Therefore, we invite you to submit a revised version of the manuscript that addresses the points raised during the review process. 

After careful consideration, we feel that it has satisfied our scientific requirements for publication.

1) However, our editorial team have significant concerns about the grammar, usage, and overall readability of the manuscript. PLOS ONE requires that published manuscripts use language which is 'clear, correct, and unambiguous', see our criteria for publication at https://journals.plos.org/plosone/s/criteria-for-publication#loc-5. We therefore request that you revise the text to fix the grammatical errors and improve the overall readability of the text.

We suggest you have a fluent English-language speaker thoroughly copyedit your manuscript for language usage, spelling, and grammar. If you do not know anyone who can do this, you may wish to consider employing a professional scientific editing service.

Please note that we will not be able to proceed with publication of your manuscript until the concerns above are addressed.

2) We note that one or more reviewers has recommended that you cite specific previously published works in an earlier round of revision. As always, we recommend that you please review and evaluate the requested works to determine whether they are relevant and should be cited. It is not a requirement to cite these works and you may remove them before the manuscript proceeds to publication. We appreciate your attention to this request.

* A copy of your manuscript showing your changes by either highlighting them or using track changes (uploaded as a supporting information file)

* A clean copy of the edited manuscript (uploaded as the new manuscript file)

A rebuttal letter that responds to each point raised by the academic editor and reviewer(s). You should upload this letter as a separate file labeled 'Response to Reviewers'. A marked-up copy of your manuscript that highlights changes made to the original version. You should upload this as a separate file labeled 'Revised Manuscript with Track Changes'. An unmarked version of your revised paper without tracked changes. You should upload this as a separate file labeled 'Manuscript'.

If applicable, we recommend that you deposit your laboratory protocols in protocols.io to enhance the reproducibility of your results. Protocols.io assigns your protocol its own identifier (DOI) so that it can be cited independently in the future. For instructions see: https://journals.plos.org/plosone/s/submissionguidelines#loc-laboratory-protocols. Additionally, PLOS ONE offers an option for publishing peer-reviewed Lab Protocol articles, which describe protocols hosted on protocols.io. Read more information on sharing protocols at https://plos.org/protocols?utm_medium=editorialemail&utm_source=authorletters&utm_campaign=protocols. We look forward to receiving your revised manuscript. 

Kind regards,

Hanna Landenmark

Staff Editor

PLOS ONE

on behalf of 

Mohit Bajaj

Journal Requirements:

Reviewers' comments:

Reviewer's Responses to Questions

**Comments to the Author**

1. If the authors have adequately addressed your comments raised in a previous round of review and you feel that this manuscript is now acceptable for publication, you may indicate that here to bypass the “Comments to the Author” section, enter your conflict of interest statement in the “Confidential to Editor” section, and submit your "Accept" recommendation.

Reviewer #1: All comments have been addressed

Reviewer #2: (No Response)

Reviewer #3: All comments have been addressed

2. Is the manuscript technically sound, and do the data support the conclusions?

Reviewer #1: Yes

Reviewer #2: No

Reviewer #3: Yes

3. Has the statistical analysis been performed appropriately and rigorously? 

Reviewer #1: Yes

Reviewer #2: (No Response)

Reviewer #3: Yes

4. Have the authors made all data underlying the findings in their manuscript fully available?

Reviewer #1: No

Reviewer #2: (No Response)

Reviewer #3: Yes

5. Is the manuscript presented in an intelligible fashion and written in standard English?

Reviewer #1: Yes

Reviewer #2: No

Reviewer #3: Yes

6. Review Comments to the Author

Reviewer #1: All my concerns have been well answered by the authors and the article has improved overall, I have no other comment.

Reviewer #2: my comments have not been addressed and I would like to reject this manuscript. I believe that author should try more to improve this manuscript.

Reviewer #3: The edits provided by the author respond to all comments.

7. PLOS authors have the option to publish the peer review history of their article (what does this mean?). If published, this will include your full peer review and any attached files.

Reviewer #1: No

Reviewer #2: No

Reviewer #3: No

---

## [Author Response · Author response to Decision Letter 1]

12 Aug 2024

Dear Editor,

Thank you so much for sending the reviewers comments related to manuscript. We applied the reviewers’ comments to improve the quality of the paper.

• Based on this, the grammatical points and writing errors of the article have been corrected in general.

• Also, the relevant references have been explained more carefully and modified in the article.

---

## [Editor Report · Decision Letter 2]

23 Aug 2024

Simultaneous Costs Minimizing in Electricity and Gas Micro-grids with the Presence of Distributed Generation

PONE-D-24-09657R2

Dear Dr. Kharrati,

We’re pleased to inform you that your manuscript has been judged scientifically suitable for publication and will be formally accepted for publication once it meets all outstanding technical requirements.

Kind regards,

Dr. Mohit Bajaj

Academic Editor

PLOS ONE
---

## [Editor Report · Acceptance letter]

9 Sep 2024

PONE-D-24-09657R2 

PLOS ONE

Dear Dr. Kharrati, 

I'm pleased to inform you that your manuscript has been deemed suitable for publication in PLOS ONE. Congratulations! Your manuscript is now being handed over to our production team.

Kind regards, 

on behalf of

Dr. Mohit Bajaj 

Academic Editor

PLOS ONE